# Measurement of Functional Use in Upper Extremity Prosthetic Devices Using Wearable Sensors and Machine Learning

**DOI:** 10.3390/s23063111

**Published:** 2023-03-14

**Authors:** Elaine M. Bochniewicz, Geoff Emmer, Alexander W. Dromerick, Jessica Barth, Peter S. Lum

**Affiliations:** 1The MITRE Corporation, McLean, VA 22102, USA; 2Department of Biomedical Engineering, Catholic University of America, Washington, DC 20064, USA; 3Medstar National Rehabilitation Network, Washington, DC 20010, USA; 4Veterans Affairs Medical Center, Providence, RI 02908, USA; 5Department of Rehabilitation Medicine, Georgetown University, Washington, DC 20057, USA

**Keywords:** machine learning, amputation, upper extremity, functional use, body-worn sensors, outcome measures, rehabilitation

## Abstract

Trials for therapies after an upper limb amputation (ULA) require a focus on the real-world use of the upper limb prosthesis. In this paper, we extend a novel method for identifying upper extremity functional and nonfunctional use to a new patient population: upper limb amputees. We videotaped five amputees and 10 controls performing a series of minimally structured activities while wearing sensors on both wrists that measured linear acceleration and angular velocity. The video data was annotated to provide ground truth for annotating the sensor data. Two different analysis methods were used: one that used fixed-size data chunks to create features to train a Random Forest classifier and one that used variable-size data chunks. For the amputees, the fixed-size data chunk method yielded good results, with 82.7% median accuracy (range of 79.3–85.8) on the 10-fold cross-validation intra-subject test and 69.8% in the leave-one-out inter-subject test (range of 61.4–72.8). The variable-size data method did not improve classifier accuracy compared to the fixed-size method. Our method shows promise for inexpensive and objective quantification of functional upper extremity (UE) use in amputees and furthers the case for use of this method in assessing the impact of UE rehabilitative treatments.

## 1. Introduction

A major goal of upper extremity (UE) post-amputation rehabilitation is to increase the use of the prosthetic limb both in and outside the patient’s home [1,2,3]. Currently, clinicians and researchers have no objective, quantitative, low-cost, and practical method to measure real-world functional use (FU) of that limb sensitively and specifically outside a laboratory setting [4,5,6,7]. We define functional use as movements that contribute to the performance of activities of daily living (i.e., reaching to grasp, gesturing, balancing functions), and nonfunctional movements are associated with gait and whole-body movements. Thus, clinicians and researchers are unable to measure the real-world impact of rehabilitation therapies on the use of the prosthetic following upper limb amputation (ULA). Given the high levels of prosthesis rejection, rehabilitation programs that focus on achieving maximum user independence can represent a key factor in increasing prosthesis acceptance and use, thus maximizing a patient’s ability to complete activities of daily living (ADLs) [8,9,10,11,12,13,14].

To evaluate changes in upper extremity functional use, clinicians and researchers have few options to utilize. A common approach is the use of in-clinic, performance-based measures to rate a patient’s ability to perform motor tasks explicitly connected with functional use [1,15,16,17,18,19,20,21]. Additionally, self-report scales are used to quantify the amount and quality of UE functional use [10,22,23,24,25,26,27]. Neither method quantitatively measures UE use in the patient’s home or community environment. Further, neither method directly evaluates the use of the UE to accomplish the principal goal of rehabilitative treatment: the use of the UE in ADLs, work-related, and leisure activities. It is simply assumed that the tests or methods are acceptable proxies for determining everyday functional prosthetic use [3,28,29].

Sensor technology is rapidly becoming mature enough to contribute to a solution for quantitative out-of-clinic, community-based assessment. A growing body of research focuses on the use of various types of sensors, such as Inertial Measurement Units (IMUs) [2,20,29,30,31,32,33,34,35,36,37]. Research of this type usually involves placing IMUs on multiple body parts and using the sensor information to infer the performance of a specific predefined activity [31,37,38,39,40]. A limitation of this method is that patients find it problematic to put on many sensors, and the sensors can be noticeable, possibly leading to limited user acceptance or changes in behavior. Identifying an inconspicuous and highly usable solution has proven difficult.

Recent results have shown that IMUs and machine learning classifiers are very effective in identifying activities in healthy controls [41,42,43,44,45,46,47,48,49,50]. The accuracies often rely on the number of IMUs used and their location, with the wrist, chest, ankle, and thigh commonly used. Accuracies higher than 90% are even possible from cell phone IMU data for detecting several classes of activities (i.e., walking, sitting, standing, jogging, upstairs, and downstairs) [46]. However, the detection of activities in patient populations is complicated by larger variability in movement patterns across the population. For example, automatic detection of vestibular gait is possible with good accuracy if data is restricted to a specific task [51]. For upper extremity tasks, several studies have found that wrist-worn IMUs and Random Forest Classifiers are superior to other processing methods for detecting functional arm use in stroke [52,53]. Another group is working to provide a more detailed description of the upper extremity function, by first identifying five primitives of the upper extremity function: reaching, transporting, repositioning, stabilizing, and idling [54]. They then used nine IMUs and a deep learning data pipeline (PrimSeq) to count the number of functional primitives performed during ADL practice and demonstrated superior performance to competing algorithms [55]. Thus, very detailed descriptions of upper extremity movements are possible with IMUs and machine learning, which promises to improve the treatment and assessment of patients with upper extremity impairment [Schambra] [56].

In prior work, we designed a sensor system and a classifier model that can be employed outside a lab setting for persons with unilateral UE impairments [20,35]. The hardware was lightweight and small enough to minimize behavioral changes and the software was designed for use on a smartphone. We previously showed that IMUs coupled with machine learning algorithms can separate functional UE movements (e.g., pushing open a door) from non-functional UE movements (e.g., arm swing with gait) in chronic stroke patients [20]. We now extend our work by presenting a method for using a single wrist-worn sensor to measure UE functional use during instrumental ADLs (IADLs) in healthy controls and persons with an upper-limb, below-elbow amputation. We also develop and test a new method of calculating data features using variable block sizes. It is unknown if prior methods that were shown to be successful in healthy controls and patients with stroke will translate into accurate functional activity identification in individuals with upper limb amputation who use a prosthetic device.

## 2. Materials and Methods

The MedStar Health Research Institute Internal Review Board (IRB) approved the study; participants provided informed consent.

### 2.1. Subjects

Subjects were recruited by convenience sampling. Control subjects were asked to self-report any neurological or physical impairments that would preclude full study participation. The sample of UE amputee participants met the following criteria: (1) age 21 or greater; (2) below-elbow amputation that occurred >6 months prior to the study; (3) no UE injury or conditions that limited use prior to the amputation; and (4) regular use of a prosthetic limb (self-reported). Table 1 lists the demographics of the amputee subjects. All control subjects were right-hand dominant, and the amputee subjects were right-hand dominant prior to their amputation.

### 2.2. Measures

Participants’ wrists were outfitted with a prototype device (see Figure 1) containing an IMU (ADIS16400BMLZ, Analog Devices) that was sampled by a microcontroller (Arduino Pro Mini) at a rate of 200 Hz. Data were stored on the prototype using a micro SD card written via an embedded logger (Sparkfun OpenLog). The SD card can record several days of continuous data. Linear acceleration and angular velocity, in three directions each, were used for the analysis. Participants wore devices on both wrists to mitigate the possibility of subjects favoring an extremity based on the presence or absence of a sensor. Subjects were not told which arm was being recorded. For the prosthetic, the sensor was placed as close to the wrist as possible. Similar to other reports [57,58], all analyses were performed only on the non-dominant UE (controls) or affected UE (amputee participants).

Subjects underwent a baseline assessment that used the Action Research Arm Test (ARAT) to assess functional activity limitations of the UE. The ARAT uses a four-point ordinal scale on 19 items, with subscales for gross motor, grasp, grip, and pinch. It has proven reliable, valid, and responsive to change across a variety of time points in post-stroke patients [15,59] and has also been used to quantify the functional limitations of amputees when using their prosthetic device [60,61,62]. The Edinburgh Inventory was used to measure handedness [63].

### 2.3. Procedures

Performance data were collected within an artificial town at the MedStar National Rehabilitation Hospital (NRH) Independence Square^®^ facility. We used the apartment, which has a kitchen and bedroom, the grocery store, and the car. We sought to acquire a sample of data that was representative of real-world UE use, and that was controlled only enough to provide analyzable data. Following the procedures found in Bochniewicz et al. [20], participants were asked to accomplish a set of instrumental ADLs, called activities, within the artificial town in a specific order. The activities were further broken down into a set of individual tasks. In two instances, subjects were instructed to complete a task either with a single UE (gather groceries) or with both UEs (hold a large box). In the rest of the experiment, subjects chose their method of completing the tasks. The activity groupings were: (1) Doing the laundry; (2) Cleaning the kitchen; (3) Shopping for groceries (including getting in and out of the car); (4) Changing the sheets on the bed.

Subjects used no adaptive equipment or assistive devices other than their prosthetic devices. Amputee subjects were not required or asked to perform the activities with their prosthetic but were instructed to use their prosthetic limb as they would in everyday life. There was no time limit for completing the activities and subjects rested as needed. To gather data on movement that was not purposeful (i.e., nonfunctional), we engaged the participants in seated conversation and asked them to occasionally walk around the artificial town for up to ten minutes. The entire experiment was videotaped, with a focus on the subjects’ UE movements. To ensure video and sensor record synchronization, the study used light-emitting diodes (LEDs) on the sensor that flashed on a preset schedule and were visible on the video.

### 2.4. Data Analysis

For the data analysis, we used the same protocol described by Bochniewicz, et al. [20]. Three annotators manually performed the video annotation and determined the class of arm use. Each frame of the video was annotated by all three annotators. The annotators coded each frame of the video using a methodology called Functional Arm Activity Behavioral Observation System (FAABOS) [20,58]. FAABOS has four coding categories and Table 2 presents the categories, classification, and examples used during annotation. We also used a fifth FAABOS category labeled Unknown when the arm was out of view of the camera.

In instances where the annotators reached no majority decision as to the FAABOS category into which a movement fell, or the majority decision was Unknown, the data were assigned to the Unknown category and excluded from the classification analysis. This applied to less than 1.5% of the data. Once the data were annotated, we focused on classifying UE movement into two broad categories: functional use and non-functional use. To do this, the FAABOS codes of Task-related and Nontask-related functional use were combined into one category labeled Functional Use, and the codes of Non-functional use and Rest state were combined into one category labeled Non-functional Use.

To classify the data, we used a custom program, written in Java, to create functional use and non-functional use data blocks. The feature vectors (1) entropy [64,65], (2) mean, (3) variance, and (4) cross-correlation between single linear and angular sensors [66] were calculated from the sensor data and used in the classification model. In the method described by Bochniewicz et al. [20], the feature vector data blocks are created by chunking the raw data into 800-data-point blocks (4 s). We also developed a second method, where variable-length chunks of data were used to create the feature vectors (Figure 2). The new method was developed to improve the performance of the classifier, especially for the inter-subject trials. The size of the chunks was determined through a multi-step process via custom code. First, a multivariate Hotelling T^2^ test [67] was run on two adjacent 2000-point chunks of the raw data. Then a threshold filter, set at an 80% true positive rate, was applied to the resulting test statistic. If the test statistic was below the threshold then a task change was denoted, and the border point separating two consecutive blocks was defined. We chose the 80% true positive rate as the threshold after also conducting the same tests with a 70%, 75%, and 85% true positive rate. The 80% true positive rate maximized the accuracy increase in the inter-subject test results while minimizing the accuracy decrease in the intra-subject test results. If no task change was detected, a 600-point shift allowed the test method to identify a more accurate task endpoint compared to the fixed block size method. The code shifted the active data blocks 600 points within the raw data file and repeated the Hotelling T^2^ test analysis. This shifting is repeated until the next border between consecutive blocks is defined. The result of using the 600-point shift is that the task change determination occurs in 600-data-point increments (see Figure 2). After performing the analysis with a variety of data chunk sizes over 10,000 times using the Block Bootstrap with Replacement methodology [68], 2000-point chunks with a 600-point shift were found to provide the highest accuracy most often. Finally, we applied the previously referenced custom program to the raw data to create the feature vectors, based on the task change annotations. Ground truth of functional or non-functional use was determined by a simple majority, assigning whichever category had over 50% of the data in each data block. This is different from the previous fixed block method, which ignored any mixed data blocks.

The sensor data were categorized into either functional use or non-functional movements using the JavaML Random Forest classifier with its default settings [69]. The Random Forrest parameters were 100 trees max, random seed set to 123 for repeatability, no limit to the depth of the trees, no back fitting, and four features randomly selected at each split. We used the Random Forest classifier because it is known to perform well over a wide variety of data sets and has shown success on movement data in the past [20,37,40]. The data were classified in two different ways: intra-subject 10-fold cross-validation testing and inter-subject leave-one-out testing. In intra-subject tests, the training and testing datasets were drawn from the same subject. In inter-subject tests, the training dataset contained data from every subject except the subject in the test set. The patient and control groups were evaluated separately. To determine the accuracy of the variable data chunk method, each temporal feature set had an extra variable attached to it that denoted the size of the data chunk. After the classifier was run, the data chunk size of the correctly classified feature sets was summed and then divided by the sum of all the data chunks. This was carried out to allow more accurate comparisons to the fixed data chunk method.

Data were analyzed in multiple ways to evaluate the classifier’s accuracy and precision. The time each subject group needed to complete each activity group and all the activities were analyzed via the Mann-Whitney U test to determine if there were significant differences between the control and amputee participants. We used the Wilcoxon signed-rank test to determine if there were significant differences between the fixed data size vs. the variable data size methods. Summary statistics are presented as medians and ranges given the nonparametric sample.

## 3. Results

Ten healthy controls (subjects 1–10; 6 women), and 5 individuals with UE amputation (subjects 11–15; 2 women) participated. The controls had a median (range) age of 42.5 (25–66) years, and the amputee subjects were 51 (22–63) years old. It should be noted that Subject 14’s prosthetic did not properly open and close. However, this subject completed all of the activities as instructed and represents a real-life scenario encountered by this population. The results from the 10 controls using the 800-point fixed window method have been previously reported [20].

All participants completed all activities and the individual tasks therein. Table 3 details the time spent on each activity, as determined by the video annotators, rounded to the nearest minute. It also shows the percentage of time that each subject spent with his or her non-dominant arm (for the control subjects) or prosthetic limb (for the amputee subjects) in the functional use and non-functional use states. In <2% of times, each subject’s arm fell into the “Unknown” or FAABOS “Category-1” state. Overall, amputee subjects required more minutes to complete the activities as compared to the control subjects. The non-dominant limb of controls completed the activities in 19.5 (16–25) minutes, while the amputees needed 26 (20–34) minutes. This difference between groups was statistically significant (*p* = 0.013). However, the percentage of time the non-dominant and prosthetic limbs were actually performing functional movements, from examining video annotations was not significantly different between groups (*p* = 0.513), with controls and amputee subjects spending a median and range of 62.5% (55.5–71.0) and 63.5% (58.3–74.9), respectively.

For the control subjects the classification accuracy for the 800-point fixed data chunk method was 95.4% (93.2–98.6) on the intra-subject train/test classifier model and 92.3% (83.2–98.2) on the inter-subject train/test classifier model. The classification accuracy for the variable data chunks method was 85.4% (63.6–91.8) for the intra-subject model and 82.5% (75.0–89.3) for the inter-subject model. For intra-subject modeling, accuracy decreased significantly by 11.1% (4.3–30.6) when using the variable-size method (*p* = 0.005), while accuracy decreased by 9.8% (3.1–15.3) for inter-subject modeling (*p* = 0.005). For the amputee subjects the classification accuracy for the 800-point data chunks method was 82.7% (79.3–85.8) on the intra-subject train/test classifier model and 69.8% (61.4–72.8) on the inter-subject train/test classifier model. The classification accuracy for the variable data chunks was 69.9% (50.1–90.8) for the intra-subject model and 71.6% (54.2–80.2) for the inter-subject model. For amputees, statistical testing found no difference in accuracy between methods (*p* = 0.138 for intra-subject and 0.225 for inter-subject). Figure 3 and Figure 4 show box plots of the Random Forest classifier’s accuracy performance. Table 4 details the amputee results for both classification methods.

Table 5 details the amount of time spent in functional use as predicted by each method and compared to ground truth. The error magnitude in functional use estimates compared to ground truth when using a fixed data window was 6.4% (1.9–17.8) in intra-subject modeling and 3.8% (1.3–4.3) in inter-subject modeling. When using the variable window size, the error magnitude was 5.1% (1.9–25.7) in intra-subject modeling and 2.5% (0.4–6.4) in inter-subject modeling. Statistical testing found no differences in functional use estimates between methods (*p* > 0.5).

## 4. Discussion

We have tested the accuracy of a machine learning method applied to wrist-worn sensors for detecting functional use of the prosthetic limb during unstructured task performance. The fixed data chunk method we developed can distinguish with good accuracy the proportion of movements that would fall into the functional use category, especially in the intra-subject modeling. This represents a significant step toward more accurately measuring the degree to which impaired limbs are integrated into ADLs over long time periods and in unconstrained environments. These methods have the potential to improve the design of restorative trials for amputees as well as a routine clinical treatment. Similar to a majority of early-stage evaluations of prototype clinical technology, this study was limited by the size of the patient population. To establish clinical validity further research with larger patient populations is required. We would need to determine how well this methodology generalizes across UE amputees and to patients with other UE impairments. Since the study was designed to represent a wide range of functional activities and movements that would occur in a home or community environment, the classifier model should perform with comparable accuracy in a completely unconstrained environment.

Examination of the time to complete activities yielded an apparent contradiction regarding amputee performance. The percentage of time performing functional movements was the same between the amputee and control groups (62–63%), however, amputees took significantly longer to complete the activities (19.5 min for controls and 26 min for amputees). One possible explanation is that amputee movements were of comparable efficiency compared to controls, but additional time was taken between functional movements to set up the prosthesis (passive DOF) or to position objects before they could be acted on by the prosthesis. An alternative explanation is that the amputees did take longer to complete functional movements or tasks, but this was offset by a compensation strategy whereby the dominant limb took over some of the tasks normally performed by the opposite limb because of limitations in prosthetic limb performance. A more detailed analysis of the video would be needed to answer this question. It is also interesting that the ARAT scores are moderate to low, ranging from 9–33 (max = 57), while the percentage of time performing functional movements in amputees was comparable to the non-dominant limb of controls. This might be explained by the fact that a sizable component of the ARAT is fine grasping skills, requiring holding small objects such as marbles and ball bearings, but these types of skills were not part of the activities tested. Additionally, shoulder and trunk compensations are a common strategy in prosthesis users [61], and these movements are marked down on the ARAT scoring. For future studies, a specific assessment of activity performance such as the Activities Measure for Upper Limb Amputees should be used [18].

Development of the variable method was motivated by our prior work, which found that most of the classification errors with the fixed data chunk method occurred in the transitions between functional and nonfunctional movements [20]. If these transitions can be detected and used to chunk the data appropriately, theoretically these errors would be reduced. Another potential benefit to the variable chunk size method is that less data is ignored than with the fixed data chunk method. In the fixed block method, blocks that included both functional and non-functional ground truth labels (mixed blocks) were not used in the training set, while all of the data is used in the variable block method. However, the results from this study did not support the use of our variable-sized data chunk procedure, as accuracies did not improve compared to a fixed data chunk method. One possible explanation is that the variable chunk method resulted in data blocks that were 2000 points or larger. This gave the machine learning less feature space data to work with than the fixed 800-point block method, as one feature vector is calculated from each data block. This may have negatively impacted the accuracy of the machine learning. However, when focusing specifically on functional use as a percentage of the total time spent during the experiment, the two blocking methods produced similar results. This is important because the primary variable of interest for clinicians is the percentage of functional use, not algorithm accuracy.

In comparing the results of the two methods, the variable data chunk classifier had a higher variance across subjects than the fixed data chunk classifier. This could be a consequence of the variable classifier working best only for certain types of subjects. While the small sample size makes it difficult to know what the significant differences between the subjects might be, one possibility is that the variable method works better with specific types of prosthetic devices. The variable method classification of both Subjects 11 and 12 showed an accuracy drop of 5.5% or more, while classification accuracies of Subjects 13, 14, and 15 all showed increases of 7.5% or more. While differences between the two methods were not statistically significant, this could indicate that certain improvements could be achieved for certain types of patients. Subjects 13, 14, and 15 had unique differences compared to Subjects 11 and 12. Subject 13 was the only subject using a body-powered prosthetic device and Subject 14’s device would not open and close properly, meaning it was used mostly for opposition posts. Subject 15 would be considered a bilateral amputee because he only had a crude grasping ability with the right UE, but used no device, thus forcing him to use his terminal device on his left side (the side used in the analysis) for all tasks that required fine motor grasping skill. However, based on this small sample size, it is not possible to draw any conclusions regarding this point.

This work extends our technique to the amputee population. Previously we reported accuracies of 88% (intra-subject) and 70% (inter-subject) in a group of individuals with stroke performing the same protocol [20]. For the amputees, the 800-point data chunk method yielded 82.7% accuracy on the intra-subject test and 69.8% in inter-subject testing. Thus, accuracies were lower than in the stroke population but may be high enough to justify the use of this method as a clinical tool. We previously showed in a cohort of stroke subjects that this approach is far more accurate than the commonly used counts thresholding method [52]. While motorized prostheses can potentially log use automatically through sensors already built into the prostheses, the wide range of componentry and vendors used by amputees supports the development of a single sensor that can be used with all prosthesis types, including body-powered devices. Moreover, we are investigating the use of the sensors on smartwatches as a practical alternative to the custom devices we developed for this project.

## 5. Conclusions

We have developed and tested an inexpensive and potentially clinically useful method for measuring functional real-world upper extremity use in amputees. Overall, there was no advantage to the variable chunk size method over the fixed chunk size approach. The intra-subject method yields the best-performing models and captures subject-specific movement patterns that are problematic for inter-subject modeling. While this requires training a new model of each patient, we are currently researching methods for reducing the amount of training data needed and/or automating the video annotation method using deep learning algorithms to eliminate the need for human annotators. Additionally, as more prosthesis users are tested, the inter-subject models should improve.

## Figures and Tables

**Figure 1 sensors-23-03111-f001:**
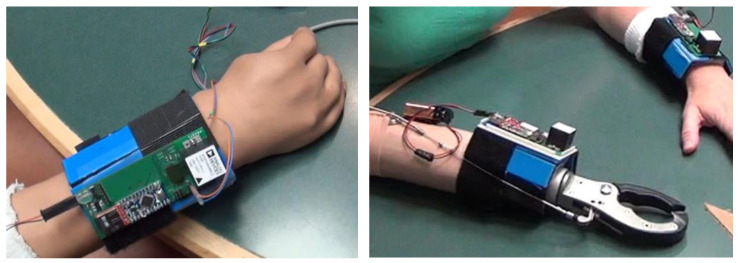
Picture of the sensors attached to a myoelectric device (**left**) and a body-powered device (**right**). The wires emanating from the myoelectric device are for synchronization purposes and are removed before starting the data collection.

**Figure 2 sensors-23-03111-f002:**
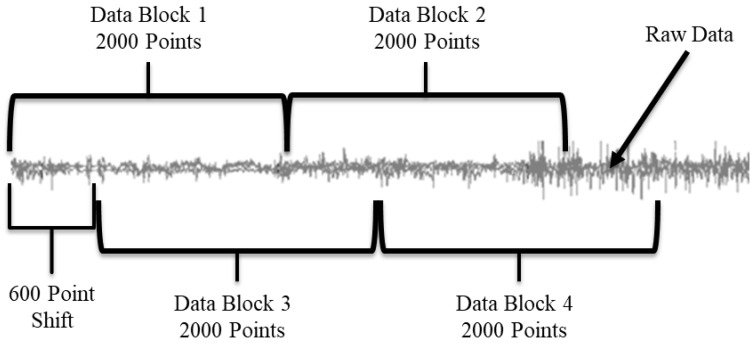
Visual representation of the Hotelling Tests application to the raw data stream. The grey line represents the raw data. Data Block 1 and Data Block 2 are compared to each other. If they are found to be different, then the first block is the first 2000 points, and Block 2 is compared to the next 2000 data points, etc. If Blocks 1 and 2 are found to be the same, then a 600-point shift is added and Blocks 3 and 4 are compared. If they are different, the first block is the first 2600 points in the data. If Blocks 3 and 4 are the same, then another 600-point shift is added, etc.

**Figure 3 sensors-23-03111-f003:**
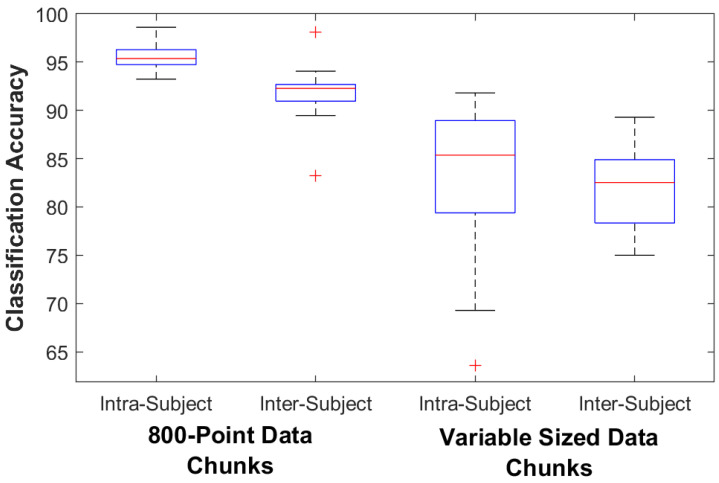
Controls Classifier Performance. The box plots show the median, 25th, and 75th percentiles. The ends of the whiskers represent the minimum and maximum of all the data. Outliers are plotted separately.

**Figure 4 sensors-23-03111-f004:**
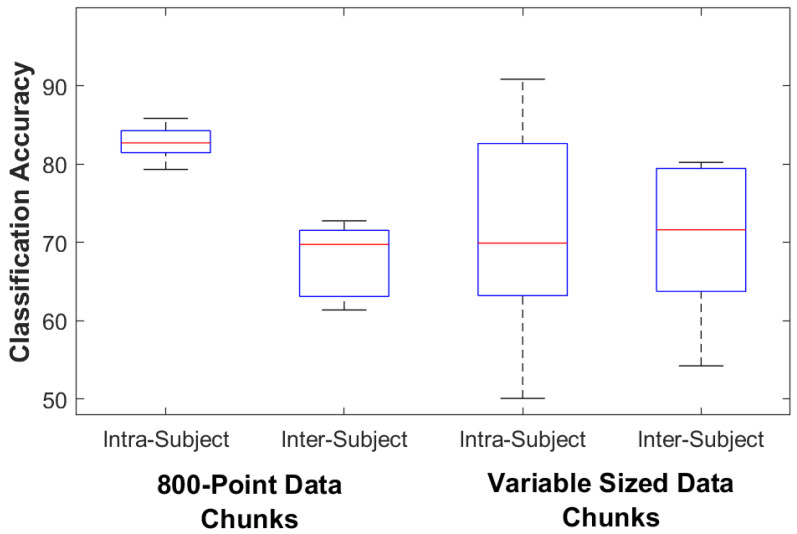
Amputee Classifier Performance. The box plots show the median, 25th, and 75th percentiles. The ends of the whiskers represent the minimum and maximum of all the data. Outliers are plotted separately.

**Table 1 sensors-23-03111-t001:** Demographics of upper limb amputee subjects.

Subject	Age	Sex	Affected UL	UE Capacity *	Prosthetic Information	
					Type	Features
11	51	M	Left	30	Myo-electric, multi-DOF	Passive wrist
12	22	F	Right	33	Myo-electric, single DOF	Three-jaw chuck, passive rotating wrist
13	38	F	Right	31	Body-powered	Split hook
14	57	M	Left	28	Myo-electric, single DOF	Three-jaw chuck, passive wrist
15	63	M	Left	9	Myo-electric, multi-DOF	Articulating fingers

* UL capacity of the affected UL was measured with the ARAT, scores are out of 57 points with higher scores indicating a more normal function of the UE. The abbreviation DOF is degrees of freedom.

**Table 2 sensors-23-03111-t002:** FAABOS categorization.

FAABOS Movement Label	Movement Description	Nonfunctional/ Functional Use Classification	Examples
Category 0	No activity or movement	Nonfunctional	Arm resting on a table during a conversation
Category 1	Nonfunctional	Nonfunctional	Arm swinging freely while walking, arm lifted/moved by another person
Category 2	Nontask-related, functional	Functional	Gesturing, covering mouth while yawning
Category 3	Task-related	Functional	Tying shoelace, pushing open a door
Category (−1)	Unknown	Excluded	NA

Abbreviation: Functional Arm Activity Behavioral Observation System (FAABOS).

**Table 3 sensors-23-03111-t003:** Time in minutes to complete the activities.

	Functional Activity Time (min)				Nonfunctional Activity Time (min) *	% Time in Functional Use
Participant	Laundry	Kitchen	Shopping	Making Bed		
*Controls*						
1	5	7	5	3	13	61.46%
2	5	6	7	4	11	55.53%
3	5	6	5	4	9	69.84%
4	6	5	4	3	10	67.92%
5	7	6	4	2	10	63.59%
6	11	5	6	3	12	70.98%
7	8	5	3	4	19	57.30%
8	5	5	3	4	14	67.49%
9	6	4	2	4	13	57.11%
10	6	5	4	4	15	58.45%
Median (range)	6 (5–11)	5 (4–7)	4 (2–7)	4 (2–4)	12.5 (9–19)	62.53% (55.5–71)
*UL Amputees*						
11	8	7	4	7	11	61.23%
12	7	6	4	5	8	70.78%
13	14	6	5	4	16	63.50%
14	6	6	4	4	11	58.33%
15	11	8	5	10	13	74.94%
Median (range)	8 (6–14)	6 (6–8)	4 (4–5)	5 (4–10)	11 (8–16)	63.5% (58.3–74.9)

* Time in minutes that participants were engaged in non-purposeful movements including walking, talking, or sitting with investigators.

**Table 4 sensors-23-03111-t004:** Amputee classifier accuracy results for the two different data segmenting methods.

Participant	Intra-		Inter-	
	800-Point Data Chunks	Variable Data Chunks	800-Point Data Chunks	Variable Data Chunks
11	85.84%	50.06%	61.36%	54.22%
12	83.73%	67.57%	72.75%	66.90%
13	82.70%	79.88%	69.75%	79.17%
14	82.17%	69.9%	63.65%	71.60%
15	79.31%	90.84%	71.13%	80.21%

**Table 5 sensors-23-03111-t005:** Percent of Time spent in Functional Use by the Amputee Subjects.

Participant	Ground Truth	Intra-Subject		Inter-Subject	
		800-Point Data Chunks	Variable Data Chunks	800-Point Data Chunks	Variable Data Chunks
11	61.23%	78.98%	86.96%	58.37%	60.87%
12	70.78%	60.76%	65.66%	75.11%	72.15%
13	63.50%	57.11%	51.02%	64.79%	66.67%
14	58.33%	52.45%	60.23%	54.52%	64.77%
15	74.94%	73.08%	71.67%	78.82%	72.41%

## Data Availability

The data presented in this study are available on request from the corresponding author. The data has not been approved for public release by MITRE Corporation.

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
