# Peer review of "Measurement of Functional Use in Upper Extremity Prosthetic Devices Using Wearable Sensors and Machine Learning"

_sensors, 2023, doi:10.3390/s23063111_

Round 1

Reviewer 1 Report

The authors present a paper about the measurement of functional use in upper extremity prosthetic devices. The paper is well-written and organized. The methods and the obtained results are clear and well argumented. Two methods are tested for data classification.

Here are some minor comments to improve the quality of the paper:

-       increase the font size of Figure 3 e 4;

-       move the bibliographic reference before the dot,

-    give more information about the Random Forest parameters used in the paper.

Author Response

1)  increase the font size of Figure 3 e 4;

Figures have been revised.

2)  move the bibliographic reference before the dot;

Corrected.

3)  give more information about the Random Forest parameters used in the paper;

The following information has been added.  “The Random Forrest parameters were 100 trees max, random seed set to 123 for repeatability, no limit to the depth of the trees, no backfitting, 4 features randomly selected at each split.”

Reviewer 2 Report

Thanks to the editors for the opportunity to review this work.

Thanks to the authors and congratulations on your effort in conducting the present research.

The title of the study was “Measurement of functional use in upper extremity prosthetic devices using wearable sensors and machine learning”.

The study was to extend a novel method for identifying upper extremity functional and nonfunctional use to a new patient population: upper limb amputees.

The strong part is that it is a relevant topic for basic research. The manuscript is clear, presented in a well-structured. Overall, you had a good paper and focused on a good point. The study can be of interest for the reader of Sensors. There are several comments:

-          The language of manuscript needs a proofreading and editing.

-          Can you please move the subjects’ details and table 2 from results to the methods section.

-          Can you please write the main conclusion of the study under section named to “Conclusion”, with note the recommendations for future work.

Author Response

1) The language of manuscript needs a proofreading and editing.

We have thoroughly proofread the paper.

2)  Can you please move the subjects’ details and table 2 from results to the methods section.

Done, and renamed to Table 1.

3) Can you please write the main conclusion of the study under section named to “Conclusion”, with note the recommendations for future work.

The following paragraph has been added as conclusions:

We have developed and tested an inexpensive and potentially clinically useful method for measuring functional real-world upper extremity use in amputees. Overall, there was no advantage to the variable chunk size method over the fixed chunk size approach. The intra-subject method yields the best-performing models and captures subject-specific movement patterns that are problematic for the inter-subject modeling. While this requires training a new model of each patient, we are currently researching methods for reducing the amount of training data needed and/or automating the video annotation method using deep learning algorithms to eliminate the need for human annotators. Additionally, as more prosthesis users are tested, the inter-subject models should improve.

Reviewer 3 Report

The study presents an interesting and constructive purpose for the improvement of rehabilitative treatments.

Overall the text is well structured as the experimental design, however there are a few suggestions that should be taken into account to improve the quality of the paper.
The Introduction lacks the definition of “functional” and “nonfunctional use” of upper limb. Moreover this paragraf should delve deeper into some arguments such as IMU literature and machine learning, particularly random forest classifier (that it is superficially motivated in Materials and Methods).

Furthermore, lines 34-37 are a bit redundant as they repeat almost the same concept.

In the Material and Methods paragraph, I suggest to add the time storage capacity of the micro SD.
The IMU device in Figure 1 doesn't seem to be small and unnoticeable. Also, were the subjects aware of which arm would be recorded?

In the aim of the study, the author described a method to measure upper extremities functional use during ADLs and instrumental ADLs while the methods described in line 112 refer only to instrumental ADL.

In line 146, what is the “custom program” to which the authors refer?

The sentence “All control subjects were right-hand dominant, and the amputee subjects were right-hand dominant prior to their amputation” on lines 201-202 should be found in Materials and methods paragraphs, specifically under Subjects.

The acronym DOF is not explained.

In Discussion section, authors start with the limitations, but I suggest to begin with an interpretation and a description of your findings' significance and then present and explain the limit of the study.

Author Response

1) The Introduction lacks the definition of “functional” and “nonfunctional use” of upper limb. Moreover this paragraph should delve deeper into some arguments such as IMU literature and machine learning, particularly random forest classifier (that it is superficially motivated in Materials and Methods).

We have added the definition of functional use in the first paragraph of the introduction: “We define functional use as movements that contribute to performance of activities of daily living (ie: reaching to grasp, gesturing, balancing functions) and nonfunctional movements are associated with gait and whole body movements.”  We have also rewritten one of the paragraphs in the introduction to include recent work on IMUs and machine learning for activity classification.

2) Furthermore, lines 34-37 are a bit redundant as they repeat almost the same concept.

Two sentences here have been combined to reduce redundancy. 

3) In the Material and Methods paragraph, I suggest to add the time storage capacity of the micro SD.

The following sentence is added: “The SD card can record several days of continuous data.”

4) The IMU device in Figure 1 doesn't seem to be small and unnoticeable. Also, were the subjects aware of which arm would be recorded?

We indicated that the device was “lightweight and small enough to minimize behavioral changes”.   We believe this is correct.  However, we do acknowledge that a smaller footprint would be preferable.  We have added a sentence to discussion: “Also, we are investigating the use of the sensors on smart watches as a practical alternative to the custom devices we developed for this project.”   Subjects were not told which arm was being recorded and in fact we recorded data from both arms.  This is indicated in methods. 

5) In the aim of the study, the author described a method to measure upper extremities functional use during ADLs and instrumental ADLs while the methods described in line 112 refer only to instrumental ADL.

Thank you for catching this error in our text and appreciating the important distinction between the two.  The study aim has been corrected to highlight that the activities in this study were instrumental activities of daily living (IADs). See the corrected text in red font at the end of the Introduction and within the procedures section of the Methods.

6) In line 146, what is the “custom program” to which the authors refer?

By “custom” program, we mean a computer program written in Java just for the purpose of the data analysis. This is now stated in the methods. 

7) The sentence “All control subjects were right-hand dominant, and the amputee subjects were right-hand dominant prior to their amputation” on lines 201-202 should be found in Materials and methods paragraphs, specifically under Subjects.

Done.

8) The acronym DOF is not explained.

Definition has been added to Table 1 footnote.  Refers to degrees of freedom on the prosthetic device used. 

9) In Discussion section, authors start with the limitations, but I suggest to begin with an interpretation and a description of your findings' significance and then present and explain the limit of the study.

This first paragraph of discussion has been flipped as suggested.  The findings and significant are first, then the limitations listed second.